# COMPRESSED-LANGUAGE MODELS FOR UNDERSTANDING COMPRESSED FILE FORMATS: A JPEG EXPLORATION

## ABSTRACT

This study investigates whether Compressed-Language Models (CLMs), *i.e.* language models operating on raw byte streams from Compressed File Formats (CFFs), can understand files compressed by CFFs. We focus on the JPEG format as a representative CFF, given its commonality and its representativeness of key concepts in compression, such as entropy coding and run-length encoding. We test if CLMs understand the JPEG format by probing their capabilities to perform along three axes: recognition of inherent file properties, handling of files with anomalies, and generation of new files. Our findings demonstrate that CLMs can effectively perform these tasks. These results suggest that CLMs can understand the semantics of compressed data when directly operating on the byte streams of files produced by CFFs. The possibility to directly operate on raw compressed files offers the promise to leverage some of their remarkable characteristics, such as their ubiquity, compactness, multi-modality and segment-nature.

## 1 INTRODUCTION

The digital world stores its information in files, and each file follows a format: a standard establishing how information is encoded into a byte structure, such as `doc` msd (2018), or `wav` Stanford University & (CCRMA). These formats allow for consistent parsing and predictable encoding of data into byte streams, independent of the specifics of the storage device. Both storage devices and bandwidth are limited, and so modern standards have defined Compressed File Formats (CFFs), *e.g.,* `mp3` of Congress (2024b), or `zip` of Congress (2024a). Unlike traditional formats, CFFs are not directly readable in their compressed state, and must undergo a decoding process to revert the data to its usable form. As a result, CFFs are complex components of our digital infrastructure, and they have become crucial for efficient storage and transmission of information.

We identify three characteristics of CFFs that are key to their widespread adoption: (i) **Ubiquity**: CFFs are universally recognized and used across both public and private digital collections, making them a standard in data storage. (ii) **Compactness**: CFFs reduce file size while preserving relevant information by exploiting statistical properties and local repetitive patterns, such as Huffman coding Huffman (1952) in JPEG Pennebaker & Mitchell (1992) and motion vectors in MPEG Puri & Eleftheriadis (1998)—for instance, one GB of compressed video expands to 50-200 GB when decompressed. (iii) **Generality**: despite the variety in compression techniques, each CFF method is capable of transforming any type of data, within a specific modality, into a streamlined one-dimensional array of bytes. These virtues of CFFs suggest the potential usefulness of models capable of directly manipulating information encoded with these formats.

A model capable of directly processing the byte arrays in CFFs could manipulate the files' semantics without needing to decompress them. Such a model, referred to as a "Compressed Model" (CM), offers significant advantages over traditional models that process uncompressed data. In particular, we note two virtues of CMs (1) **Compressed Input**: CMs would interact with compressed data, which is inherently compact and less redundant than uncompressed data. This characteristic eliminates the need for common techniques for reducing redundancy, such as pooling or attention windowing. (2) **Raw Input**: CMs could exploit vast archives of compressed data exactly as they are stored, thus

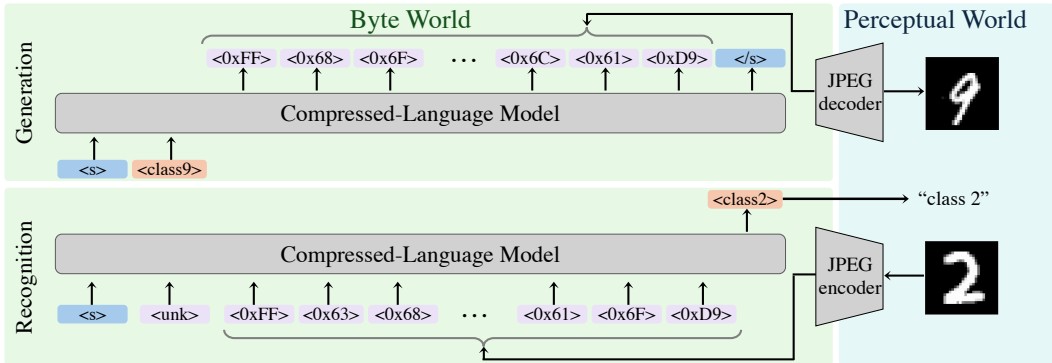

Figure 1: **Testing the understanding capacity of "Compressed-Language Models" (CLMs)**, *i.e.* language models trained for next-token prediction on byte streams produced by Compressed File Formats (CFFs). We use the JPEG format as a case study for CFFs. We test the CLMs' understanding of JPEG files by probing their capacity to generate new files, recognize real files, and handle files with anomalies (not displayed in the figure). Our evidence suggests that CLMs can understand JPEG.

bypassing the need for domain-specific pre-processing and enabling efficient data handling. While these advantages highlight the desirability of CMs, their implementation remains under-specified.

We recognize that the sequence-like property of the byte streams produced by CFFs can inform the implementation of CMs. Notably, these byte streams exhibit three language-like properties: *syntax*, established by the CFFs that produced the streams; *semantics*, inherited from the encoded information; and a *finite lexicon*, corresponding to their byte alphabet. Given these attributes, we hypothesize that an effective approach to implementing CMs is through language models Bengio et al. (2000); Bahdanau et al. (2014). We term such models "Compressed-Language Models" (CLMs), and argue that this implementation would achieve a third property of (3) **Universality**, standardizing the training pipeline: sequence-to-sequence models as the architecture, next-token prediction as the optimization objective, and CFFs as the universal language for encoding and pre-processing diverse types of information, such as images, text, video, *etc.* These facts suggest that building a CLM by applying the paradigm of language models on byte streams produced by CFFs is promising.

Yet, the feasibility of a CLM understanding the byte streams produced by CFFs remains largely unexplored Lester et al. (2024). Notably, various properties of CFFs could hinder the understanding of these streams. In particular, CFFs introduce unique complexities with their layered compression schemes, which transform the original data through multiple stages to produce the final byte stream Pennebaker & Mitchell (1992). The compact nature of compressed files also implies that even minor alterations in the byte stream or its interpretation could drastically affect the decoded information. Moreover, since existing language models were not designed considering these challenges, it remains uncertain whether they are capable of effectively understanding the "language" of CFFs.

In this study, we test the understanding capability of Compressed-Language Models, *i.e.* language models trained for next-token prediction on byte streams produced by Compressed File Formats. There is a wide variety of CFFs, and so, aiming at practicality, we focus our study on the JPEG format Wallace (1992) for encoding image data. We choose JPEG due to its widespread use and because it directly exemplifies key concepts of CFFs, such as compact representation, byte level encoding, multilayered compression, and versatility. Furthermore, JPEG files are easy to evaluate by employing the JPEG decoder and directly inspecting the resulting image raster. We test if CLMs understand JPEG files, in three standard image datasets, by probing their capabilities across three axes: recognizing file properties, handling files with anomalies, and generating new files. Please see Fig. 1 for an overview of our methodology for testing understanding. To the authors' best knowledge, this is the first study testing the understanding of language models that operate directly on CFF-encoded data. Our evidence supports the hypothesis that CLMs can understand compressed files.

## 2  PRELIMINARIES

We select the JPEG format as a case study for Compressed File Formats (CFFs) for three key reasons: commonality, representativeness and ease of evaluation. Regarding its commonality, JPEG is widely

used due to its efficient encoding and decoding algorithms, as well as its high perceptual quality, making it one of the most prevalent image formats on the internet. Regarding its representativeness, JPEG exemplifies key concepts in CFFs, including lossy compression, quantization, domain-knowledge operations (*e.g.,* discrete cosine transforms), and entropy coding, which are foundational in many compressed formats Pennebaker & Mitchell (1992); Boutell (1997); Deutsch (1996); Puri & Eleftheriadis (1998). Finally, judging the quality of generated JPEG files is straightforward by decoding the compressed data and conducting visual inspection.

Despite these virtues, JPEG may present challenges from the perspective of language modeling, as the encoding is a multi-layered procedure that produces byte streams with complex patterns and long-term dependencies. Next, we provide a brief overview of the JPEG encoding, transforming an image into a byte stream, and refer the interested reader to JPEG Committee (2024) for specifics.

**JPEG Encoding Overview.** Initially, JPEG converts the image from the RGB color space to YCbCr, separating its information into luminance and chrominance. The chrominance channels are then often down-sampled, leveraging the human eye's lower sensitivity to color differences. The image is then divided into 8×8 pixel blocks, where each block then undergoes a discrete cosine transform (DCT) to obtain a frequency-domain representation. The DCT coefficients are then scaled with a set of pre-defined quantization tables that control the level of lossy compression. The resulting coefficients are then subjected to run-length encoding to represent consecutive repeated values as a single value and a count. Finally, the symbols in the sequence are compressed with Huffman or arithmetic coding, resulting in the final byte stream. The "lossy-ness" of JPEG is tuned via a *JPEG quality* parameter, which balances between compression and perceptual quality by controlling the quantization tables.

**Potential challenges offered by JPEG.** Considering the complex encoding procedure presented above, we enlist possible challenges that JPEG poses to language models. Firstly, the compressed byte sequences produced by JPEG are highly sensitive to modifications, since a single change in the byte stream can drastically alter the resultant image. This sensitivity contrasts sharply with natural language, where redundancy is common and often less crucial to the meaning of text. Furthermore, the run-length and Huffman coding processes used in JPEG compression operate at the bit level rather than the byte level. This fact means that the boundaries between encoded symbols may not align neatly with byte boundaries, potentially resulting in encoded symbols starting or ending in the middle of a byte. This misalignment complicates the model's ability to learn from byte-level data, as it introduces difficulties in interpreting the encoded information with precision. Lastly, the interactions between the DCT, quantization, and the corresponding quantization tables introduce complex inter-dependencies among the bytes. These relationships are non-trivial and pose significant challenges for language models, which typically excel when dependencies are more reflective of patterns found in natural language.

## 3  METHODOLOGY

This section outlines our methodology for testing the hypothesis that a language model can be directly trained on data resulting from Compressed File Formats and effectively capture the underlying semantics. We propose to test the model's understanding of JPEG data by assessing its performance along three main axes: (i) Recognition of inherent properties of compressed files, (ii) Discovery and correction of anomalies in compressed files and (iii) Generation of compressed files. These tasks rely on the model's ability to interpret and manipulate the JPEG's lexicon (the byte values), syntax (the structure of the byte sequences), and semantics (the meaningful interpretation of file content), and thus reflect diverse aspects of understanding the encoded byte sequence.

We minimize the impact of training schemes and network-design choices by following the current standard practices for decoder-only language architectures and optimize for next-token prediction Brown et al. (2020). The only factor on which we diverge from these defaults is how we construct the inputs to the model, *i.e.* the tokenized "sentence" on which the model operates.

**Vocabulary and Tokenization.** The model's vocabulary consists of the 256 possible byte values, from `0x00` to `0xFF`, with additional start (``) and end (``) of sequence tokens. Such vocabulary enables the model to process a file's raw byte stream directly. That is, analogous to character-level tokenization in natural language, we represent each byte with a separate token.

**Input Sequence Construction.** To enable both class-conditional generation and recognition immediately after training, we extend the input sequences to include token indicators of both (1) JPEG quality and (2) semantic class. Specifically, each sequence or "sentence" fed to the model has both a JPEG quality and a semantic class token that surround the file's byte sequence. For instance, a JPEG file of quality 75 and class 3 is represented as:

```
<q75> <class3> <bytes> 0xFF ... 0xD9 </bytes> <q75> <class3>
```

This template guides the model both for conditional generation (conditioned on JPEG quality and class) and for regression of those attributes after scanning the byte sequence. Please refer to the appendix for an explanation of how we avoid the degenerate solution of copy-pasting these attributes.

We next describe in detail the three main axes on which we evaluate the model's understanding of CFFs, recognizing files (Sect. 3.1), handling anomalous files (Sect. 3.2), and generating new files (Sect. 3.3), and report the corresponding results in Sect. 4.

## 3.1 FILE RECOGNITION

*Can the model recognize properties of a given file?* A language model should be able to understand the attributes of sentences Collobert et al. (2011); Devlin et al. (2018); Lagler et al. (2013). Similarly, a model tuned for JPEG byte sequences should be capable of recognizing the properties of a given file. Thus, we test the model's capacity to recognize the JPEG quality and semantic class of real files.

Here, we leverage our input-sequence template for probing the model for this task. Specifically, we previously outlined how the JPEG quality and class token are (optionally, see appendix) provided at the start and end of the byte sequence (B). To evaluate the model's capability to recognize these properties from the byte sequence only, we replace the first two tokens of the sequence with the *unknown* "`<unk>`" token. That is, we represent a validation sample as: "`<unk> <unk> <bytes> *B </bytes>`" (see Fig. 1). We then use the model to auto-regressively predict the next two tokens in the sequence, corresponding to the model's predictions for B's JPEG quality and semantic class. With then use these predictions to compute the model's accuracy for predicting each property.

The model's recognition capacities are a direct consequence of the model's next-token prediction pre-training. That is, the next-token prediction objective simultaneously tasked the model with learning to generate byte sequences and recognizing them. However, in a sentence, the number of tokens contributing to the generation objective (hundreds of tokens) is disproportionately large w.r.t. the number of tokens contributing to the recognition objective (only two tokens). This lack of proportion suggests that the model's recognition capacities after pre-training is likely a lower bound to the model's actual potential. We thus study the possibility of leveraging the model's generative pre-training Brown et al. (2020), and fine-tune the model to improve its recognition capabilities. To this aim, we fine-tune the model on these same sentences, but only supervise the token on which we are interested for recognition (either the JPEG quality or the semantic class token). After this fine-tuning, we again measure the model's classification performance via accuracy.

## 3.2 FILE ANOMALY HANDLING

*Can the model handle files that contain anomalies?* A language model should be able to handle sentences with grammar errors or typos Santos et al. (2018); Al-Jefri & Mahmoud (2013); Cheng et al. (2020); Etoori et al. (2018); Ji et al. (2021). Analogously, a model that understands JPEG files should be capable of handling anomalies in these compressed files.

To test such capabilities, we consider three anomaly-related tasks: (i) tagging a file as anomalous, (ii) detecting the precise location of an anomaly in a file, and (iii) correcting a given anomaly. To study these tasks, we require a dataset of anomalous files. We choose to generate such a dataset, based on real files, via the procedure we describe next.

**Simulating Anomalous Files.** To generate a dataset of anomalous encoded files, we start by considering a collection of $M$ regular files $X = \{x_i\}_{i=1}^M$, where each file $x_i$ is a list of $N_i$ bytes, *i.e.* $x_i = [b_1, \ldots, b_{N_i}]$, and $x_i[\,k\,] = b_k \in \{0, \ldots, 255\}$. Our dataset considers files with one-byte-substitution anomalies. For each file $x_i$, we thus generate all one-token perturbed variants by modifying each byte in $x_i$ with all possible 255 values different from its original value. Formally, we

define the perturbation function $\Psi$:

$$\Psi(x, v, j) = x \oplus (x[\,k\,] \rightarrow v),$$

where $\oplus$ denotes replacing the $k^{\text{th}}$ token in $x$. By applying this function to each token, we construct the set of all possible perturbed sequences for file $x_i$ as:

$$\hat{X}_i = \{\Psi(x_i, v, k) \mid v \in \{0, \ldots, 255\} \setminus \{x_i[\,k\,]\},\ k = 1, \ldots, N_i\}, \tag{1}$$

where $v$ thus considers every byte value except the original one. This operation yields $255 \times N_i$ anomalous variants of each file $x_i$. We aggregate this set across all files and obtain a comprehensive dataset of anomalous files $\hat{X} = \bigcup_{i=1}^{M} \hat{X}_i$. This dataset provides a testbed for a thorough evaluation of the model's sensitivity to one-byte-substitution anomalies. Next, we detail the three tasks on which we evaluate the model's capacity to handle anomalies.

**Task #1: Tagging.** *Can the model tag files as anomalous?* The ability of the model to distinguish between normal and anomalous files can be assessed by examining its likelihood estimates. Specifically, the model should assign higher likelihoods to "correct" or "natural" files and lower likelihoods to those that are anomalous.

We formalize our testing approach. For each original file represented by the byte sequence $x_i$ in our dataset, we consider all its perturbed variants. In particular, we hypothesize that the model assigns a higher likelihood to the original file $x_i$ than to any of its perturbed variants, denoted by $\hat{x}_{i,j} \in \hat{X}_i$. Formally, we denote the log-likelihood of a file $x_i$ by $L(x_i)$, as computed by the model, and define the difference in log-likelihoods between the original sequence $x_i$ and one of its perturbed variants as $\Delta L_{i,j} = L(x_i) - L(\hat{x}_{i,j})$. Our hypothesis thus poses that $\Delta L_{i,j}$ should be positive, indicating that, according to the model, original files are more likely than their corresponding anomalous variants. We statistically test this hypothesis by applying the Wilcoxon Signed-Rank Test Woolson (2007) to the set of log-likelihood differences between all files $x_i$ and their corresponding $255 \times N_i$ anomalous versions, which we denote as $\mathcal{L} = \{\Delta L_{i,j} \mid x_i \in X,\ \hat{x}_{i,j} \in \hat{X}_i\}$.

In this test, the null hypothesis $H_0$ poses that the median of these differences is less than or equal to zero, suggesting no significant difference in likelihoods between natural and perturbed files. Conversely, the alternative hypothesis $H_1$ argues that the median is greater than zero, supporting the model's capability to identify anomalies. Formally,

$$H_0 : \text{median}(\mathcal{L}) \leq 0 \quad versus \quad H_1 : \text{median}(\mathcal{L}) > 0.$$

We evaluate this hypothesis using a significance level of $\alpha = 0.05$. That is, we reject the null hypothesis if the resulting $p$-value from the Wilcoxon Signed-Rank Test is less than $\alpha$, supporting our conjecture that the model effectively discriminates between natural and anomalous files.

**Task #2: Detection.** *Can the model identify the location of an anomaly?* We study if, given an anomalous file, the model is capable of identifying the exact location of the byte causing the anomaly in the file. This task is analogous to how a language model should be able to identify the exact location of a typo in a sentence.

For this purpose we consider the model's capacity for estimating the likelihood of individual tokens in the sequence. Let $\hat{x}^k$ denote an anomalous sequence whose $k^{\text{th}}$ token was perturbed. Then, we hypothesize that the model assigns a particularly low likelihood (*i.e.* a high perplexity) to the anomalous token. Formally, we suggest the model can predict the location $k$ of the anomaly by simply sorting the estimated likelihoods of the individual tokens, that is:

$$\hat{k} = \arg\min_l L\left(\hat{x}^k[\,l\,] \mid \hat{x}^k[\,:l-1\,]\right) \tag{2}$$

Thus, we use the model's per-token likelihood estimates to identify the anomaly in each sequence in the dataset $\hat{X}$. We then interpret the task as a simple classification problem and compare each prediction $\hat{k}$ with the ground truth $k$ to compute classification accuracy.

**Task #3: Correction.** *Can the model correct an anomalous file?* We study if, given the exact location of an anomalous byte in a file, the model is able to replace such byte with the correct one. This task is analogous to how a language model should be able to fix a typo in a sentence by replacing the wrong character with the right one.

We propose a correction process that uses the model's likelihood estimates to replace the anomalous token with the most likely one. Specifically, assume the $k^{\text{th}}$ token in a file sequence $\hat{x}^k$ is anomalous. We thus propose to correct the sequence by simply substituting the anomalous token with the one that maximizes that token's likelihood, conditioned on the preceding tokens. This approach hypothesizes that the model, trained for next-token prediction in files, naturally suggests the *correct* token when given the preceding context. Formally, we define the correction operation as follows:

$$\hat{x}_{\text{corrected}} = \hat{x}^k \oplus \left( \hat{x}^k[\, k\,] \to \arg\max_b L\left(b \mid \hat{x}^k[\,:k-1\,]\right)\right),$$

where $\hat{x}^k[\, k\,]$ refers to the anomalous byte value, and $\arg\max_b L\left(b \mid \hat{x}^k[\,:k-1\,]\right)$ denotes the byte that the model considers to be most likely at position $k$, given the tokens at all previous positions.

We apply this correction method to each anomalous file in our dataset of perturbed files $\hat{X}$. Following our evaluation of *Task #2: Detection*, we interpret the output of the correction task as a classification problem, and compare each $\hat{x}_{\text{corrected}}$ with the ground-truth $x$ to compute classification accuracy.

### 3.3 FILE GENERATION

*Can the model generate new files?* A language model should be able to generate new sentences Lagler et al. (2013); Floridi & Chiriatti (2020); Team et al. (2024); Touvron et al. (2023). Analogously, a model that understands JPEG files should be capable of generating novel files that adhere to the JPEG standard. We thus test the model's capacity to generate new JPEG files.

We perform class-conditional generation by feeding the model a prompt stating the quality and semantic class of the target file (*e.g.,* "`<q30> <class0> <bytes>`") and then auto-regressively generate the file content. We continue the generation process until the model generates the `</bytes>` delimiter, after which we write the generated bytes to a file with the `.jpeg` extension.

**Checks.** File validity: The generated file may not represent a *valid* JPEG file: it may include malformed headers, incorrect byte sequences, or inconsistencies in the compression structure. To check the file's validity, we use the standard OpenCV library to attempt opening the file, and consider the generation invalid if the library either throws an error or a warning when attempting to open the file. We measure the model's performance as the percentage of generated sequences that correspond to valid JPEG files. JPEG quality: For files that pass the validity check, we assess the JPEG quality presented by the byte stream. To check the file's JPEG quality, we use the ImageMagick library, which determines the compression quality based on the quantization tables in the file's header. We measure the model's performance as the percentage of the valid files for which the file's exhibited JPEG quality matches the quality token specified in the prompt.

## 4 EXPERIMENTAL RESULTS

Our methodology aims at testing if a model trained for next-token prediction on JPEG data understands the information underlying the encoded data. In this section, we first describe implementation details of our experiments, and then report results on each of the three axes we consider for testing the model's understanding of JPEG, as described in our Methodology (Sect. 3).

### 4.1 IMPLEMENTATION DETAILS

**Data.** Our study tests if the standard practices for training a sequence model, when applied directly on JPEG-encoded data, result in a model that understands the information underlying the compressed data. We thus focus on the encoding rather than on the data, and so minimize confounding factors stemming from complex data by experimenting on two simple image datasets: MNIST LeCun (1998) and CIFAR-10 Krizhevsky et al. (2009). On the one hand, MNIST consists of 60K training and 10K test grayscale images of 10 handwritten digits, providing a simple and well-defined testbed. Its straightforward nature is ideal for isolating challenges related to the JPEG encoding process without the interference of complex image characteristics. On the other hand, CIFAR contains 50K training images and 10K test color images across 10 classes, offering a richer diversity in textures and subjects. This variability introduces more complexity, which allows us to assess how the model processes more intricate content when encoded in JPEG format. To enhance the model's generalization, we

Table 1: **Anomaly Detection: accuracies for locating anomalies in JPEG files.** The model displays a notable capacity for identifying the precise location of anomalies.

| Top-k | MNIST | | | CIFAR | | |
|---|---|---|---|---|---|---|
| | Broken | Valid | Overall | Broken | Valid | Overall |
| 1 | 97.3 | 82.0 | 84.4 | 91.8 | 77.7 | 80.2 |
| 3 | 99.8 | 92.7 | 93.7 | 99.7 | 85.2 | 87.9 |
| 5 | 99.9 | 94.6 | 95.4 | 99.8 | 87.5 | 89.8 |

Table 2: **Anomaly Correction: accuracies for correcting bytes in anomalous JPEG files.** The model achieves remarkable capacity for correcting files, especially when they are broken.

| Top-k | MNIST | | | CIFAR | | |
|---|---|---|---|---|---|---|
| | Broken | Valid | Overall | Broken | Valid | Overall |
| 1 | 100 | 72.9 | 76.1 | 100 | 74.2 | 77.7 |
| 3 | 100 | 84.5 | 86.4 | 100 | 78.3 | 81.2 |
| 5 | 100 | 88.9 | 90.3 | 100 | 81.0 | 83.6 |

perform data augmentations in the image space and then save the augmented images in JPEG format. In particular, for MNIST, we apply small rotations. For CIFAR, we perform crops and horizontal flips. We also experiment with TinyImagenet Le & Yang (2015) to explore the ability of CLMs to deal with larger datasets and file sizes, we present these results in the appendix.

**JPEG quality parameters.** We resize the images to $32 \times 32$ pixels and save them as JPEGs with various quality parameters $q$. In particular we experiment with nine values, $q \in Q = \{30, 50, 60, 70, 75, 80, 85, 90, 92\}$, which are common in GIMP, Photoshop, `libjpeg`, *etc.*

**Model.** We use a small LLaMA-like model Touvron et al. (2023); parameters are in the appendix.

## 4.2 File Recognition

**Performance before and after fine-tuning.** *Before:* We find that the next-token prediction training yields a model that naturally exhibits high recognition capacity. In particular, on MNIST's validation set, we find that the model's accuracy for recognizing a file's semantic class is $97.1 \pm 0.05\%$ (average and standard deviation computed across three runs), while its accuracy for recognizing the file's JPEG quality is $100 \pm 0\%$. On CIFAR, the model reaches a more conservative accuracy of $56.9 \pm 0.4\%$ for the semantic class; however, the model still reaches an accuracy of $100\%$ for recognizing JPEG quality. *After:* The models' performance at recognizing JPEG quality is outstanding, while recognizing the semantic class has room for improvement. We thus fine-tune the models for recognizing the semantic class. After fine-tuning, accuracy on MNIST grows from $97\%$ to above $99\%$, while the accuracy on CIFAR improves more dramatically, from $57\%$ to over $74\%$.

Conclusion: We find that generative pre-training results in perfect recognition of JPEG quality on both datasets; recognition of the semantic class is high on MNIST, though lower on CIFAR. We also find that performance at semantic classification can easily be boosted via simple fine-tuning.

## 4.3 File Anomaly Handling

**Anomalous Files Dataset.** We simulate anomalous files for both MNIST and CIFAR following the procedure described in Sect. 3.2. For this procedure, we only consider 10 files (one per class) for each dataset, given the computational and storage costs of the combinatorial space of even one-byte-substitution anomalies. We find that this procedure yields both "broken" and "valid" files. That is, OpenCV recognizes some of these files to be valid JPEG files, while recognizing others as invalid. For MNIST, $15\%$ of the anomalous files are broken, while the remaining $85\%$ are valid. On CIFAR, these percentages are, correspondingly, $18\%$ and $82\%$. We next report the results for each of the three anomaly-related tasks we described in Sect. 3.2.

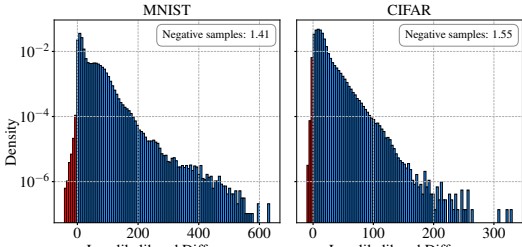

**Task #1: Tagging.** We report the histogram of log-likelihood differences in Fig. 2. For both datasets, we observe that the vast majority of differences are positive, indicating that the model assigns significantly higher likelihoods to the original sequences than to their perturbed counterparts. Running the Wilcoxon Signed-Rank Test further supports the intuition from the his-

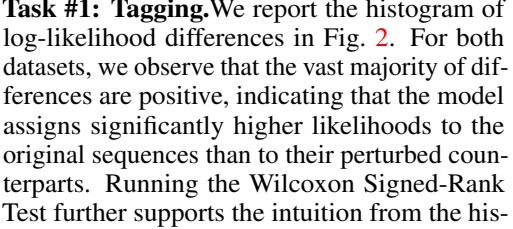

Figure 2: **Histograms of log-likelihood differences, $\mathcal{L}$, for MNIST and CIFAR.** Positive values dominate, indicating that the model consistently assigns higher likelihoods to natural files compared to their perturbed counterparts. This fact allows the model to correctly tag anomalous files.

togram. In particular, the test yields a $W$ statistic of $> 10^{11}$ for MNIST and of $> 10^{12}$ for CIFAR, with their associated $p$-values being numerically zero, and thus rejecting $H_0$ as described in Sect. 3.2.

Conclusion: our findings suggest that the model's file-level likelihood estimates are extremely sensitive to the presence of single-token anomalies in JPEG files. That is, we find that the model is capable of tagging files as anomalous even with anomalies in a single byte.

**Task #2: Detection.** Our findings from *Task #1:* suggest that the model's likelihood can distinguish when files have even single-token anomalies. Such finding suggests the model may be sensitive not just at the file level but also at the finer token level. We thus measure the performance of the anomaly detector from Eq. equation 2 at various prediction chances $k \in \{1, 3, 5\}$, and report results in Tab. 1.

For MNIST, we note that the overall accuracy at top-1 is about 85%, surpasses 90% at top-3, and reaches over 95% at top-5. The case of anomalous but valid files follows a similar trend. Notably, for the case of broken files, the detector's performance is remarkably high: the top-1 prediction is correct over 95% of times, and this number reaches almost 100% when considering the top-3 predictions. This high performance in broken JPEG files is largely expected: a file breaks when its strict format is perturbed, and deviations from that strict format are most likely easily detected by the model. For CIFAR, we mostly observe a similar trend to that of MNIST, but with reduced values: overall performance at top-1 is 80% and reaches 90% at top-5. Performance in broken files starts at above 90% and reaches essentially 100% when considering more prediction chances. Finally, on CIFAR's valid files, performance starts at 78% at top-1, and reaches 87% at top-5.

Conclusion: the model's token-level likelihood estimates are useful for identifying the precise location of anomalous bytes in JPEGs. This finding is particularly sound for broken files, *i.e.* for files in which the perturbation broke the file's strict format.

**Task #3: Correction.** Our findings from *Task #2: Detection* suggest that the model has the capacity to locate an anomaly in a file's byte sequence, and so we next explore whether it can also correct these anomalies. We report the performance on the correction task in Tab. 2.

For MNIST, we observe that the model performs the correction task exceptionally well, with top-1 correction accuracy achieving 100% for broken files and 76% overall. The performance increases notably with more prediction chances, reaching over 90% at top-5. CIFAR exhibits a similar pattern, albeit with slightly lower accuracy: starting at 78% for top-1 and climbing to almost 84% for top-5. This evidence demonstrates the model's capability to rectify anomalies, even more effectively in broken files where the format deviations are clear-cut.

Conclusion: The model performs remarkably well in the correction task, particularly when dealing with severely corrupted, *i.e.* broken, JPEG files.

## 4.4 File Generation

We generate files by sampling from the model with greedy decoding. We thus generate $90 = 10 \times 9$ files for each dataset, *i.e.* for the 10 semantic classes of MNIST/CIFAR and the 9 JPEG quality values we considered. For MNIST, we find that 99% of the files we generate, *i.e.* all except one file, (i) are valid JPEG files, and (ii) have JPEG quality matching the one in the corresponding prompt. In CIFAR, the percentage of valid JPEG files lowers slightly to 97%, *i.e.* all except three files, while all the valid JPEGs have a quality matching the corresponding prompt. Thus, in both datasets, essentially all sampled files are valid JPEGs, except for only four samples (one in MNIST and three in CIFAR). The warnings raised by OpenCV when trying to open these files mention issues such as the presence of extraneous bytes, premature ends of data segments, and wrong Huffman tables. In general, we find that sampling from the model with greedy decoding, in general, yields files that adhere to the JPEG format and also exhibit the correct JPEG quality parameter that was stated in the prompt.

We check the visual properties of the generated samples by decoding the JPEG files into image arrays. We report a subset of these samples, including both positive and negative (*i.e.* corrupt) results, in Fig. 3. We make three observations from these qualitative results: (1) Corrupt files in both MNIST (quality 85 and class "2") and CIFAR (quality 75 and class "1") are still decodable by OpenCV, and exhibit strongly-structured errors that mirror JPEG's $8 \times 8$-block standard. For the broken MNIST image, for example, sampling is successful in generating over half of the blocks but, after a certain block (in raster order), the sample displays checkerboard patterns and color shifts. (2) All samples lack strong

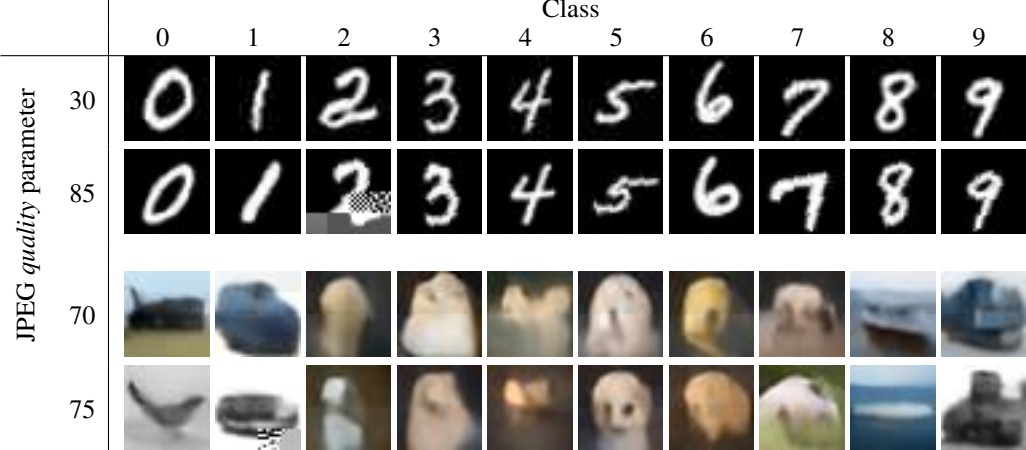

Figure 3: **Qualitative results of file generation.** We sample files from the model, via greedy decoding, and use the JPEG decoder to obtain an image raster. Here we report images of various JPEG qualities (30, 70, 75, and 85) and all semantic classes from both MNIST (first two rows) and CIFAR (bottom two rows). The bottom row of each dataset displays one sample that was recognized by OpenCV to be a corrupt JPEG. The offending samples are noticeable by their block-like artifacts (MNIST: quality 85, class "2"; CIFAR: quality 75, class "1").

artifacts except the corrupt files. That is, all samples are positive results w.r.t. visual quality except for a single one. (3) On the one hand, essentially all MNIST samples display the right semantic class as provided in the prompt; on the other hand, on CIFAR, the model seems to suffer from mode collapse in various classes. We report samples from beam search in the appendix.

## 5 RELATED WORK

**Models trained on partially-decoded JPEGs.** Instead of operating on RGB pixel values of images, a few works have devised Convolutional Neural Network (CNN) Gueguen et al. (2018); Verma et al. (2018) or Vision Transformer (ViT) Park & Johnson (2023) architectures that operate on partially decoded JPEG images. For instance, Gueguen et al. (2018) introduced a CNN LeCun et al. (1995) that processes these images, allowing for faster image handling by omitting some decoding stages. Similarly, Park & Johnson (2023) applied a ViT Dosovitskiy et al. (2020) to partially decoded JPEGs, enhancing the ability of models to interact with data in a more compressed state. These approaches mark a significant shift towards using the structural properties inherent in compressed formats, albeit still involving some level of decoding.

**Byte Sequence Modeling for Uncompressed Data.** Transitioning to the modeling of byte sequences, significant innovations have been introduced to manage the complexities associated with raw data streams. The MegaByte framework Yu et al. (2024), for instance, features a multi-scale decoder transformer architecture that addresses the challenges posed by long byte sequences through a novel "byte patchification" technique. Concurrently, bGPT Wu et al. (2024) leverages a decoder-only Transformer model for autoregressive generation of byte sequences. These methods underscore developments in processing byte sequences, yet they focus on uncompressed data, neglecting the more common and practical compressed file formats. In contrast, our work tackles the challenge of processing compressed formats, specifically using JPEG as a case study.

**Direct Operations on Compressed JPEG Byte Sequences.** ByteFormer Horton et al. (2023) ventured into processing compressed file formats at the byte level, and revealed that JPEG byte sequences introduce significant challenges due to their nonlinear encoding and variable length. This work showed that traditional byte patching methods could degrade performance, owing to the high density of information in compressed formats. Unlike ByteFormer, which employs a tailored encoder architecture for byte sequences, our work uses a straightforward decoder-only Transformer language model trained with the vanilla next-token prediction objective. We find that, despite the simplicity of this design choices, this vanilla Transformer effectively works on JPEG sequences for multiple tasks.

## 6 CONCLUSIONS

In this study, we found evidence suggesting that Compressed-Language Models (CLMs) can effectively understand JPEG-encoded byte streams. In particular, we found CLMs exhibit abilities in file recognition, anomaly handling, and file generation, without requiring decompression. Our findings pave the way for future developments in efficient data processing techniques that directly operate on files encoded by Compressed File Formats, or even segments of these files.

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

# SUPPLEMENTARY MATERIAL

## A    LIMITATIONS

Our study is limited by its focus on JPEG-encoded MNIST and CIFAR datasets, which may not represent the complexity of more varied real-world images. This restriction could impact the generalizability of our findings, as the characteristics and challenges of JPEG compression in more varied and natural images are not fully explored.

## B    TRAINING SENTENCES

In Sect 3, we described the general template for our sentences as

```
<q75> <class3> <bytes> 0xFF ... 0xD9 </bytes> <q75> <class3>
```

In practice, we introduce additional delimiters for the condition, but omit them here for clarity. Furthermore, notice that we introduced condition tokens (JPEG quality and semantic class) both before and after the segment corresponding to the file's bytes. On the one hand, the condition tokens before the file allow the model to condition its output on a specific condition, *i.e.* conditional generation. On the other hand, the condition token after the file instruct the model to scan the entire file and then predict the file properties corresponding to the condition. Unfortunately, the model may learn to solve the recognition task by copying the condition token provided before the file and pasting it after the file. To prevent this degenerate solution, we stochastically feed the model with either a "generation" or a "recognition" version of the sample during training. In the generation version, we remove supervision for the condition tokens after the file, so the model lacks incentives to copy-paste. In the recognition version, we restore that supervision, but replace the condition tokens before the file with an `<unk>` token, preventing copy-pasting and thus forcing the model to rely on the file to predict the class.

## C    CLMs on ImagenetTiny

To test the CLMs ability to deal with larger datasets and larger file sizes. We conducted experiments using Tiny Imagenet Le & Yang (2015). This dataset is a subset of ImageNet Deng et al. (2009); Russakovsky et al. (2015) that contains 100k images from 200 classes (500 for each class) downsized to 64×64. We follow use the same setup we used in our original methodology. It's worth noting that this images are double the size of CIFAR and MNIST, thus the computation required to pre-train and fine-tune on this dataset is substantially increased. Initially we trained with the original TinyImagenet split *without any data augmentations*, the pre-trained model achieved a 19.7% accuracy on the validation split. Then, we decided to augment the training set of TinyImagenet with images from the original Imagenet dataset that belong to the same class, this augmented training set highly benefited the pre-training, helping it achieve a 29.8% on the TinyImagenet validation set. This result shows that CLMs follow the usual properties of next-token prediction objective and scale well when more data is available. Furthermore, we fine-tuned the pre-trained CLM on classification only, this model achieved a competitive 34.9% accuracy.

We pre-trained during 48 hours using 1 A100 GPU, and fine-tune for an extra 24 hours on the same machine. It's also worth noting that we did not see a loss saturation during pre-training indicating that the results can be further improved if more computation is available. However, we leave this scaling up to future works with larger computational resources.

## D    Model Training

**Architecture and training objective.**    We use a decoder-only Transformer architecture Vaswani et al. (2017). The training objective is to predict the next token in a sequence, which aligns with the typical setup for auto-regressive language models The next-token prediction objective involves training the model to predict the probability distribution of the next token given the preceding sequence:

$$\max_\theta \; \mathbb{E}_{x \sim D} \left[ \sum_{t=1}^{n} \log p_\theta \left( x[t] \mid x[: t-1] \right) \right]. \tag{3}$$

## E    Generation: sampling with beam search

In Sect. 4.4 we found that sampling from the model with greedy decoding generated valid and reasonable JPEG files. We thus also experiment with beam search sampling. For sampling sensibly-looking JPEGs with beam search, we find important to constrain sampling to consider only the smallest set of tokens whose accumulated probabilities surpass a given hyper-parameter.[1] We report samples from beam search in Fig. 4.

## F    Implementation Details

**Learning hyper-parameters**    We base our implementation on the `llama-recipes` code-base Meta LLaMA (2024), and so follow their defaults when unspecified. We report training hyper-parameters in Tab. 3.

**Computer Resources.**    We conduct all our trainings in an A100 GPU of 80 GB. We run training on MNIST for 24 hours, and on CIFAR for 72. The experiment on MNIST shows signs of convergence, while the one on CIFAR does not.

Testing a trained model for recognition requires less than 10 minutes on either dataset. Sampling an MNIST file requires around 5 seconds, while sampling a CIFAR files requires almost 9, due to the difference in sequence length.

---

[1]This constraint is enforced via beam search's `top_p` hyper-parameter in the HuggingFace library.

| Configuration | Value |
|---|---|
| Precision | `bfloat16` |
| Optimizer | AdamW |
| Optimizer momentum | $\beta_1, \beta_2 = 0.9, 0.999$ |
| Weight decay | 0.0 |
| Learning rate | $7 \times 10^{-4} \,|\, 6 \times 10^{-4}$ |
| Learning rate schedule | cosine decay |
| Warm-up iterations | 10 | 11 |
| Total epochs | 6 | 5 |
| Batch size | 32 | 16 |

Table 3: **Training hyper-parameters.** The parameters follow the template "(MNIST) | (CIFAR)"

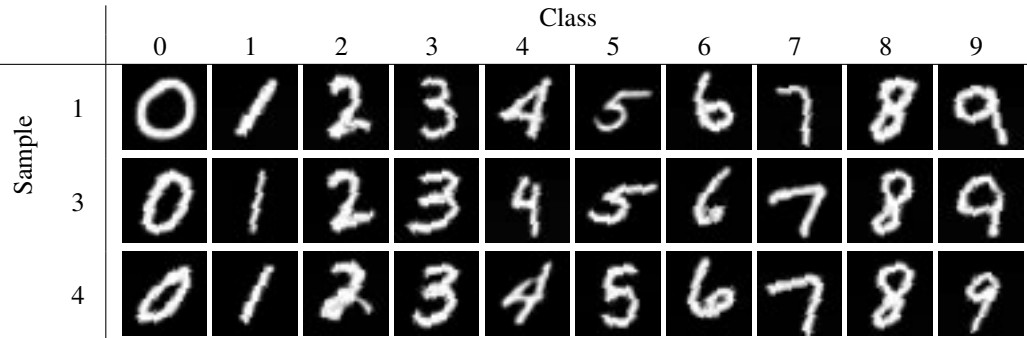

Figure 4: **Qualitative results of file generation via beam search.** We sample files from the model via beam search and use the JPEG decoder to obtain an image raster.

