# OpenReview forum: "Compressed-Language Models for Understanding Compressed File Formats: a JPEG Exploration"
_ICLR.cc/2025/Conference — ICLR 2025 Conference Withdrawn Submission_

### Official Review · Reviewer_Gj5i · 2024-10-31

**Soundness:** 2
**Presentation:** 3
**Contribution:** 1
**Rating:** 3
**Confidence:** 5

**Summary:**

Experimental study that trains and evaluates a decoder-only Transformer model directly on JPEG byte streams (= JPEG-compressed images). Training is done autoregressively as usual. Evaluation tasks are (i) predict JPEG image quality setting and class of example, (ii) detect/locate/fix single-byte errors, (iii) data generation. The key takeaway is that this works reasonably well.

**Strengths:**

S1. The exploration of LLMs to directly handle compressed data is interesting. Even if it may not turn out to be of notable relevance in practice, it may help to shed more insights into the limitations of LLM.

S2. The evaluation tasks are reasonable first steps and shed some light into the compression/decompression capabilities of plain LLMs.

**Weaknesses:**

W1. Paper positioning not convincing. The paper is motivated by a large argument that directly working on compressed format is beneficial. These arguments, however, are inherently flawed. First, arguments about ubiquity, compactness and generality are invalid because if one simply decompresses before training the LLM, these advantages would still hold. Second, while I do see the worth of studying compressed file formats (see S1), I fail to see practical relevance. On the one hand, real-world machine learning pipelines do consist of many domain-specific techniques; e.g., data augmentation (e.g., crop/scale images), helpful tokenization (e.g., SentencePiece), task-specific training objectives (e.g., BERT training) or models (e.g., CNNs). On the other hand, spending resources to "teach" a model to decompress/compress when we actually know how to do this more efficiently (JPEG encoder/decoder) is a waste of resources.

W2. Training and evaluation setup not convincing. For task (i), the prediction targets of image quality and class are fed into the training process in a somewhat contrived way to deal with problems of decoder-only models for this task. It's not clear why a decoder-only LLM is the right approach in the first place. For task (ii), the authors make "erroneous bytes" are less likely than "correct bytes" arguments. But when doing so, they ignore the entire input after the erroneous token (for localization/correction). This, again, a consequence of using decoder-only models. For task (iii), the automatic check is solely on file validity, but ignores the quality of the generated samples (other than the anecdotal examples of Fig. 3).

W3. Limited insight. This is for two main reasons. First, the papers make broad claims about compressed file formats, but then only considers JPEG, includes JPEG-specific information into the training pipeline (quality setting), and use only one image size. Second, the paper puts too much focus on whether tasks (i)-(iii) work reasonably well with an out-of-the-box LLM training pipeline. What's much more interesting, however, is exploring where such approaches would fail and why.

**Questions:**

None

---

### Official Review · Reviewer_8CVH · 2024-11-03

**Soundness:** 1
**Presentation:** 2
**Contribution:** 1
**Rating:** 3
**Confidence:** 4

**Summary:**

The paper studies whether language models trained on compressed file format can be used on three tasks on JPEG files, including recognition of file properties, handling anomalies and generating new files.

**Strengths:**

Evaluating language model's capability on JPEG byte stream is interesting. It is a bold idea with a potential to show unseen capability or to reveal important limitations of language models.

**Weaknesses:**

1. Models trained on text and models trained on compressed data have significantly different token space. While the token space of text is demonstrated learnable with various language models, the binary streams produced by compression algorithms may not have generic patterns that are generalisable for a variety of compressed data. It has been argued that the data distribution properties may be the key that drives in-context learning in transformers[1]. I feel that a detailed examination of the token distribution in the compressed data should be provided to justify the approach.

[1] Chan, S., Santoro, A., Lampinen, A., Wang, J., Singh, A., Richemond, P., McClelland, J. and Hill, F., 2022. Data distributional properties drive emergent in-context learning in transformers. Advances in Neural Information Processing Systems, 35, pp.18878-18891.

2. The experiments are done on images with very small dimensions (28x28 for MNIST and 32x32 for CIFAR, additional experiment in appendix with an image dimension of 64x64 ). It is not a surprise that a large model can fit the small search space and provide predicting and generative capability on these datasets.

3. It is likely the language model is tuned to overfit a set of small images. This is little evidence based on the technical presentation of this paper that the model has learned the format of JPEG, therefore the method is unlikely to generalise to data in compressed file formats.

**Questions:**

1. File anomaly handling (section 3.2) only considers one-token (one-byte) perturbation.  How realistic such anomaly exists in real world applications and results in actual problems?

2. line 361, "For this procedure, we only consider 10 files (one per class) for each dataset". How do the 10 files produce a result that "15% of the anomalous files are broken"? Did I miss anything?

---

### Official Review · Reviewer_hFMP · 2024-11-04

**Soundness:** 2
**Presentation:** 3
**Contribution:** 2
**Rating:** 3
**Confidence:** 3

**Summary:**

This paper investigates whether Compressed-Language Models (CLMs) can understand files compressed in the JPEG format. The authors test the models' capabilities in three aspects: recognition of file properties, handling of anomalous files, and generation of new files.
The study uses simple image datasets (MNIST and CIFAR-10) presented in encoded format as sequence data to train a small LLaMA-like model to conduct the experiment. The results suggest the model can effectively perform these tasks without decompression.

**Strengths:**

The research topic is novel as it explores the understanding capabilities of language models on compressed file formats, specifically focusing on JPEG. This area has potential for applications in efficient data storage and retrieval.

**Weaknesses:**

The paper claim that the focus of the research is on testing the understanding capabilities of compressed language models (CFFs). Besides, they draw the conclusion that “CLMs can understand the semantics of compressed data”. But the test is conduct on only one model trained by the author, instead of any existing language models. Therefore, I think the result is insufficient to support the conclusions drawn in the paper.
 It looks that JPEG-encoded formats exhibit language-like properties like any other sequences and the object is also to optimize for next-token prediction. It is not that surprising to see a model trained on sequence data works well within the same datasets (CIFAR-10 and MINIST, though as encoded data). Therefore, I think the main finding of this paper lacks some novelty.
The paper outlines the characteristics of compressed file formats (CFF) and the challenges compressed language models (CLMs) encounter when meeting CFFs but does not clarify enough the need for CLMs to address CFFs or the importance of testing their understanding capabilities.

**Questions:**

1. Why there is only single-byte replacement anomalies considered when simulating anomalous files, would other types of anomalies have a more significant impact on model performance?

2.   It seems the results of Section 4.2 are not presented in any table or figure in this section. does the term “MNIST’s validation set” refer to the validation set used during training? If so, what are the results on the test set? Could you clarify how the dataset was actually split?

3.  What’s the detail of fine-tune the models for recognizing the semantic class.? What was the data used for fine-tuning like?

---

### Official Review · Reviewer_XJ3s · 2024-11-09

**Soundness:** 3
**Presentation:** 2
**Contribution:** 2
**Rating:** 5
**Confidence:** 4

**Summary:**

The main goal of this project is to study the effectiveness of Compressed-Language Models (CLMs) in understanding raw byte streams from compressed file formats (CFFs). Specifically, they have used JPEG data in this study and evaluated the performance of CLMs on three functions: identifying inherent properties of compressed files, discovery of anomalies in compressed files, and generation of compressed files.

**Strengths:**

The authors have investigated the effectiveness of CLMs in handling raw byte streams from compressed files. Specifically, they have used the JPEG format as the compression mechanism. They have employed three datasets: MNIST, CIFAR-10, and TinyImagenet. The models used are standard models available in the literature (e.g., a small LLaMA-like model). Their tokenization is somewhat new. In general, the results indicate that CLMs are good in dealing with compressed data.

For instance, the accuracy they obtain for file recognition are: 99% on MIST and 74% on CIFAR. The model seems to be very effective in anomalies detection and files generation. For example, in the context of MNIST, 99% of the files generated are valid JPEG files.

**Weaknesses:**

The novelty of the work is modest.

**Questions:**

None

---

### Comment · Area_Chair_3BQK · 2024-11-13
**authors - reviewers discussion open until November 26 at 11:59pm AoE**

Dear authors & reviewers,

The reviews for the paper should be now visible to both authors and reviewers. The discussion is open until November 26 at 11:59pm AoE.

Your AC

---

### Note · Authors · 2024-11-24

I have read and agree with the venue's withdrawal policy on behalf of myself and my co-authors.